# Efficient Two-Step Parametrization of a Control-Oriented Zero-Dimensional Polymer Electrolyte Membrane Fuel Cell Model Based on Measured Stack Data

Zhang Peng Du [1],*[ID], Christoph Steindl [2][ID] and Stefan Jakubek [1]

[1] Institute of Mechanics and Mechatronics, Technische Universität Wien, Getreidemarkt 9, 1060 Vienna, Austria; stefan.jakubek@tuwien.ac.at

[2] Institute of Powertrains and Automotive Technology, Technische Universität Wien, Getreidemarkt 9, 1060 Vienna, Austria; christoph.steindl@tuwien.ac.at

* Correspondence: zhang.peng.du@tuwien.ac.at; Tel.: +43-1-58801-325540

**Abstract:** This paper proposes a new efficient two-step method for parametrizing control-oriented zero-dimensional physical polymer electrolyte membrane fuel cell (PEMFC) models with measured stack data. Parametrizations of these models are computationally intensive due to the numerous unknown parameters and the typically nonlinear, stiff model properties. This work reduces an existing model to decrease its stiffness for accelerated numerical simulations. Subdividing the parametrization into two consecutive subproblems (thermodynamic and electrochemical ones) reduces the solution space significantly. A parameter sensitivity analysis further reduces each sub-solution space by excluding non-significant parameters. The method results in an efficient parametrization process. The two-step approach minimizes each sub-solution space's dimension by two-thirds, respectively three-fourths, compared to the global one. An achieved $R^2$ value between simulation and measurement of 91% on average provides the required accuracy for control-oriented models.

**Keywords:** polymer electrolyte membrane fuel cell; control-oriented model; grey-box modeling; analytical differentiability; model reduction; parameter sensitivity analysis; fisher information; efficient parameterization; data-driven identification; transient operation measurement data

## 1. Introduction

PEMFCs are promising candidates for replacing internal combustion engines in mobile applications. However, fuel cell (FC) driven vehicles are still far from having a significant market share because of various challenges. One of the challenges is to control the FC during transient operations, especially regarding avoiding harmful operating conditions. Another one is to improve the FC system's efficiency via optimal control. Moreover, experimental expenditures during development are high but are reducible by using simulations instead. One of the first steps towards resolving the named and other challenges is obtaining a proper FC model. PEMFCs are promising candidates because they offer the rapid startup and low operating temperature required for automotive applications. However, it requires appropriate water management to address liquid water formation. Solid oxide FCs use non-noble metal catalysts, which results in low raw material costs, but the high operating temperature leads to thermal stress and precludes this FC for transient applications. Molten carbonate FCs can reform a wide variety of fuel sources, but the long startup time eliminates this FC for anything but continuous-power applications. Phosphoric acid FCs provide easy water management, but the low power density is not appropriate for portable applications. Alkaline FCs have the highest demonstrated operating efficiency of any FC system, but the intolerance to carbon dioxide forces the use of carbon dioxide removal equipment and pure oxygen and hydrogen [1].

In general, modeling approaches are distinguishable into three groups. First, black-box modeling means fitting an artificial model to replicate the measured input and output data

correlation [2–4]. The model structure and parameters do not need to have any physical meaning, and extrapolation capabilities are limited. Black-box models, in general, do not replicate internal unmeasured physical states. Second, white-box models solely describe the FC with first principles known from theory. All model parameters are physical quantities, fundamental constants, and known values. Measurement data is not needed [5–7]. This approach is not feasible if not every parameter is known, which is usually the case for FC models. Third, grey-box models combine both approaches. They use the first principles known from theory and require measurement data to determine the model structure and parameters. This work does not consider black-box models because the internal states' knowledge is essential for the controller to avoid harmful operating conditions. White-box models are unsuitable because many parameters are unknown. Therefore the combination of both approaches, grey-box models, provides the needed structure and fidelity for FC control. Control-oriented grey-box PEMFC models aim to have a low spatial dimension to be numerically efficient for real-time applications [8–12]. Parametrizing grey-box PEMFC models is not straightforward because they are nonlinear, numerically stiff, and have numerous unknown parameters. This work proposes an efficient two-step parametrization method that drastically simplifies the optimization problem. The method subdivides the FC model into two submodels for parametrization, each yielding a lower-dimensional sub-solution space compared to the one of the entire model.

The standard parametrization procedure determines as many parameters as possible from theory, datasheets, and expert knowledge. The remaining unknown parameters have to be obtainable by fitting measurement data. For example, McKay et al. [13] developed a lumped parameter model, and they identified the tunable parameters using least squares. Unfortunately, this approach yields a high dimensional solution space for models with many unknown parameters. The proposed multiple-step method by Xu et al. [14] estimates the nozzle coefficients first, and in the following step, a nonlinear least-squares algorithm identifies the electrochemical parameters for a steady-state voltage response. Additionally, a fitted neural network describes the dependency of the electrochemical parameters on the operating conditions. Using a neural network is suboptimal because the applicability is highly limited to the range of the available measurement data. Müller et al. [15] conducted an approximated sensitivity analysis by varying the signals and parameters. The sensitivity analysis's sole purpose was to show the importance of accurate sensors for parametrization. They used specific measurements to identify parameters one by one. Curve fitting leads to the remaining parameters. Individually estimating the parameters is not always feasible because it is strongly dependent on the available measurement data and model structure. Moreover, incorporating the sensitivity analysis into the parametrization process will increase its efficiency, especially for large-scale problems. Goshtasbi et al. [16] approximated the parameter sensitivities via difference quotients. In [17], the sensitivity analysis determines the most sensitive parameters for identification, and a global optimizer identifies this subset of parameters altogether. This approach's drawback is that a large subset still leads to a computationally-intensive problem. In [18], the authors used the parameter sensitivity analysis to divide the parameters into three groups. The groups contain the identifiable parameters in the low, medium, and high current regions, respectively. Unfortunately, this approach needs appropriately designed experiments. Ritzberger et al. [9] developed a PEMFC model focusing on its analytical differentiability. Existing models are usually not analytical differentiable but are transformable to a model with this property. Analytic differentiability enables an accurate and efficient calculation of the derivatives compared to a numerical approximation and is advantageous for various control applications. They locally linearized the nonlinear model in multiple operating points. Their proposed parameterization method optimizes each analytically linearized model for each operating point dependent on the same parameter set simultaneously. In [10], they additionally conducted a parameter sensitivity analysis based on the analytically linearized models. This approach's disadvantage is that it is limited to data with small excitations because of

the linearization. Therefore it requires specially designed experiments, which is mostly not the case for given data.

In this work, the two-step parameterization method is demonstrated with the model developed by Ritzberger et al. [10] because of its advantageous property of analytical differentiability. The model is adapted to reduce its stiffness for more efficient numerical evaluations. Subdividing the model into a thermodynamic and an electrochemical submodel leads to a parameterization method with two consecutive steps. In the first step, the thermodynamic submodel, which consists of a system of first-order ordinary differential equations (ODEs), is parametrized. The electrochemical submodel, which does not have any ODEs, is parametrized in the following second step. The subdivision reduces the solution space's dimension for each submodel, and an analytic parameter sensitivity analysis further reduces each parameter subset by excluding non-significant parameters. A global optimizer identifies the parameter subsets by minimizing the difference between simulated and measured output signals [19]. An existing FC test bench delivers the measurement data. Compared to the available approaches, this efficient method simplifies the parametrization problem, it does not require appropriately designed experiments, and the parameter sensitivity analysis is analytically evaluable.

This paper is subsequently structured as follows: Section 2 describes the PEMFC model and its reduction. Section 3 presents the two-step parametrization method, explains the parameter sensitivity analysis, and shows the proposed method's validation. Section 4 depicts the experimental setup, and finally, Section 5 discusses the identification results.

## 2. Fuel Cell Model

This section briefly portrays the used PEMFC model to demonstrate the proposed two-step parametrization method. Additionally, this section describes the model reduction for accelerated numerical simulations. The proposed method is, of course, not limited to the described FC model. Any model separable into a submodel with ODEs and a submodel without ODEs is utilizable. The model does not have to be analytically differentiable because the parameter sensitivity analysis is numerically approximable. However, numerical approximations are less efficient and accurate evaluable than analytical solutions.

### 2.1. Model Description

Ritzberger et al. [10] developed the model, and it is an adapted version of the Pukrushpan et al. [8] model. The difference is that the adapted model has the property of analytical differentiability, which is the reason for its selection. This property is beneficial for various control methodologies [20,21], and parameter sensitivity analyses (Section 3.2). The model is a zero-dimensional physical PEMFC model, and Figure 1 gives a schematic overview. It utilizes, amongst others, mass balances, linear nozzle equations, diffusion equations, electrochemical equations, and the ideal gas law [22–24]. However, it does not consider energy balances and thus cannot model the FC temperature over time. The model treats the measured temperature as an input. This temperature is assumed to be the uniform temperature for the whole FC. The model consists of a cathode, anode, membrane, and electrochemical submodel, which are interconnected. The model equations and their derivations are not the focus of this work. More information in this regard is available in [10].

The cathode submodel has four mass states (oxygen mass $m_{ca,O_2}$, nitrogen mass $m_{ca,N_2}$, vapor mass $m_{ca,vap}$, and liquid water mass $m_{ca,liq}$) and two pressure states (supply manifold pressure $p_{ca,sm}$, and exit manifold pressure $p_{ca,em}$). The anode submodel also has four mass states (hydrogen mass $m_{an,H_2}$, nitrogen mass $m_{an,N_2}$, vapor mass $m_{an,vap}$, and liquid water mass $m_{an,liq}$) but only one pressure state (exit manifold pressure $p_{an,em}$). The membrane submodel only has the membrane water activity state $a_m$. The model inputs are the air mass flow $\dot{m}_{ca,in}$, the hydrogen mass flow $\dot{m}_{an,in}$, the anode supply manifold pressure $p_{an,sm}$, the relative humidity of the air mass flow $\varphi_{ca,in}$, the purging signal $\alpha_{purge}$, the FC temperature $T$, the atmospheric pressure $p_{atm}$, and the current $I$. The outputs are the

cathode supply manifold pressure $p_{ca,sm}$, the cathode exit manifold pressure $p_{ca,em}$, the anode exit manifold pressure $p_{an,em}$, and the voltage $U$. Hence, the following equations describe the nonlinear FC state-space model:

$$\dot{\mathbf{x}}_{nr} = \mathbf{f}_{nr}(\mathbf{x}_{nr}, \mathbf{u}, \theta) \tag{1}$$

$$\mathbf{y} = \mathbf{g}_{nr}(\mathbf{x}_{nr}, \mathbf{u}, \theta) \tag{2}$$

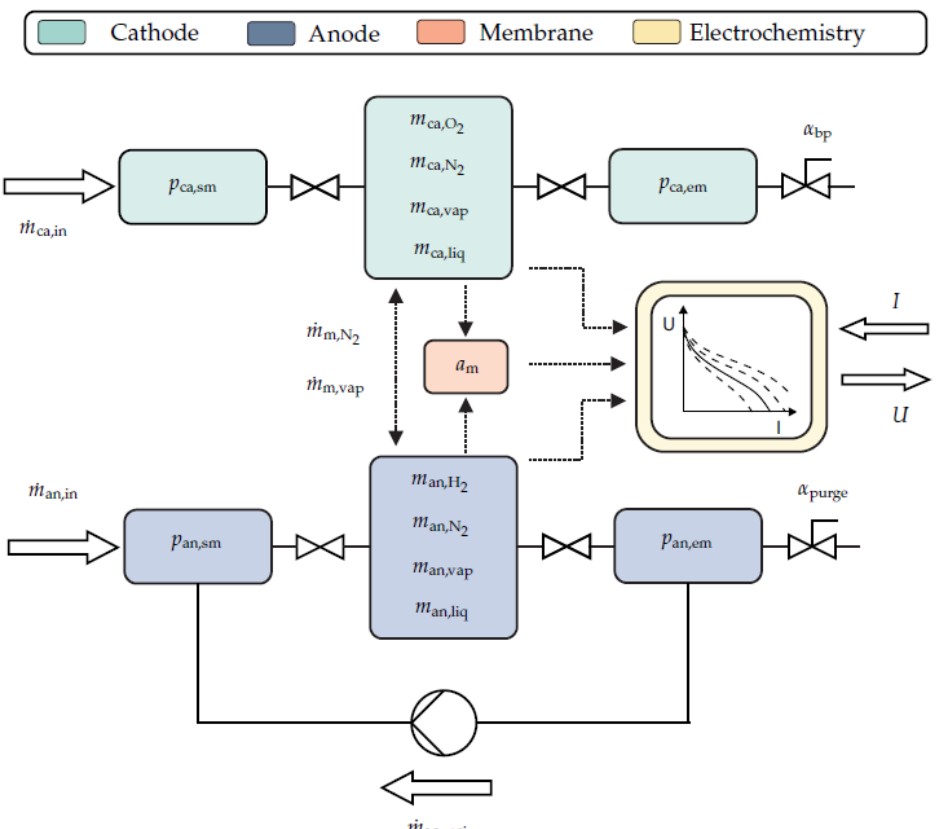

**Figure 1.** Schematic overview of the model structure of the lumped, transient FC stack model [10].

Here, $\mathbf{x}_{nr} = \mathbf{x}_{nr}(t)$ denotes the non-reduced state vector, $\mathbf{u} = \mathbf{u}(t)$ the input vector, $\mathbf{y} = \mathbf{y}(t)$ the output vector, $\theta \in \mathbb{R}^{25}$ the parameter vector, $\mathbf{f}_{nr}$ the non-reduced system function, $\mathbf{g}_{nr}$ the non-reduced output function, and $t$ the time [25]. The respective vectors are structured as follows:

$$\mathbf{x}_{nr} = \begin{bmatrix} p_{ca,sm} \\ m_{ca,O_2} \\ m_{ca,N_2} \\ m_{ca,vap} \\ m_{ca,liq} \\ p_{ca,em} \\ m_{an,H_2} \\ m_{an,N_2} \\ m_{an,vap} \\ m_{an,liq} \\ p_{an,em} \\ a_m \end{bmatrix}, \mathbf{u} = \begin{bmatrix} \dot{m}_{ca,in} \\ \dot{m}_{an,in} \\ p_{an,sm} \\ \varphi_{ca,in} \\ \alpha_{purge} \\ T \\ p_{atm} \\ I \end{bmatrix}, \mathbf{y} = \begin{bmatrix} p_{ca,sm} \\ p_{ca,em} \\ p_{an,em} \\ U \end{bmatrix}, \theta = \begin{bmatrix} V_{ca,sm}, V_{ca,cm}, V_{ca,em}, \cdots \\ \cdots V_{an,sm}, V_{an,cm}, V_{an,em}, \cdots \\ \cdots k_{perm}, k_{cond}, k_{evap}, \cdots \\ \cdots k_{ca,sm,out}, k_{ca,em,in}, \cdots \\ \cdots k_{ca,em,out}, k_{an,sm,out}, \cdots \\ \cdots k_{an,em,in}, k_{an,em,out}, \cdots \\ \cdots k_{an,leak}, \tau_m, \cdots \\ \cdots \epsilon_2, R_c, E_{ca,act}, E_{an,act}, \cdots \\ \cdots K_{ca}, K_{an}, CD_{ca}, CD_{an} \end{bmatrix}^T \tag{3}$$

$V$ denotes the volumes, $k$ the nozzle or mass flow coefficients, $\tau_m$ the water activity time constant, $\epsilon_2$ the membrane conductivity parameter, $R_c$ the ohmic contact resistance, $E$

the energy, $K$ the intrinsic exchange current parameter, and $CD$ the combined diffusion coefficient. Compared to the model described by Ritzberger et al. [10], this one has an additional output, the cathode exit manifold pressure $p_{ca,em}$. It is measured, therefore considering it as a supplementary output has the advantage that it yields additional insight into the system. Furthermore, the pressure difference between the exit manifold and the environment is too big to be sufficiently described by a linear nozzle equation. Hence nonlinear nozzle equations (derived from the Bernoulli equation) replace the linear ones at each exit manifold.

### 2.2. Model Reduction

The model reduction goal in this work is to reduce the model's stiffness for accelerated numerical simulations. Lambert [26] stated that stiffness occurs when (numerical) stability requirements, rather than those of accuracy, constrain the (simulation time) step length. The pressure dynamics of the model are faster by multiple orders of magnitude compared to the mass dynamics. Thus the pressure dynamics are assumed to be steady-state at all times. The ODE for the cathode supply manifold pressure $p_{ca,sm}$ results from a combination of the ideal gas law and the mass balance

$$\dot{p}_{ca,sm} = \frac{R_{ca,sm}T}{V_{ca,sm}}(\dot{m}_{ca,in} - \dot{m}_{ca,sm,cm}), \tag{4}$$

where $R_{ca,sm}$ denotes the cathode supply manifold's mass-specific gas constant, $\dot{m}_{ca,sm,cm} = k_{ca,sm,out}(p_{ca,sm} - p_{ca,cm})$ the mass flow between the cathode supply and center manifold, $k_{ca,sm,out}$ the cathode supply manifold's outflow nozzle coefficient, and $p_{ca,cm}$ the cathode center manifold pressure. Assuming steady-state, the ODE (4) transforms into

$$p_{ca,sm} = \frac{\dot{m}_{ca,in}}{k_{ca,sm,out}} + p_{ca,cm}. \tag{5}$$

The exit manifold pressures of the cathode $p_{ca,em}$ and anode $p_{an,em}$ are addressed in a similar fashion leading to the reduced nonlinear FC state-space model given by

$$\dot{\mathbf{x}} = \mathbf{f}(\mathbf{x}, \mathbf{u}, \theta), \tag{6}$$

$$\mathbf{y} = \mathbf{g}(\mathbf{x}, \mathbf{u}, \theta). \tag{7}$$

Here, $\mathbf{f}$ denotes the reduced system function, $\mathbf{g}$ the reduced output function, and $\mathbf{x} = \mathbf{x}(t)$ the reduced state vector. The reduced state vector is structured as

$$\mathbf{x} = [m_{ca,O_2}, m_{ca,N_2}, m_{ca,vap}, m_{ca,liq}, m_{an,H_2}, m_{an,N_2}, m_{an,vap}, m_{an,liq}, a_m]^T, \tag{8}$$

and it does not contain the three pressure states ($p_{ca,sm}$, $p_{ca,em}$, and $p_{an,em}$) anymore. The input vector $\mathbf{u}$, the output vector $\mathbf{y}$, and the parameter vector $\theta$ remain unchanged. The longest integration step length of the reduced model, which still yields a stable solution, is about 50% longer than for the non-reduced model, leading to a roughly 36% shorter simulation time [27].

## 3. Two-Step Parametrization Method

### 3.1. Key Idea

The two-step parametrization method's key idea is to subdivide the given nested state-space model (6) and (7) into two submodels. The nested model is separable in the following way:

$$\mathbf{x} = \mathbf{x}_{\text{th}}, \qquad\qquad \mathbf{f}(\mathbf{x}, \mathbf{u}, \theta) = \mathbf{f}_{\text{th}}(\mathbf{x}, \mathbf{u}, \theta_{\text{th}}) \tag{9}$$

$$\mathbf{y} = \begin{bmatrix} \mathbf{y}_{\text{th}} \\ y_{\text{el}} \end{bmatrix} = \begin{bmatrix} p_{\text{ca,sm}} \\ p_{\text{ca,em}} \\ p_{\text{an,em}} \\ \hline U \end{bmatrix}, \qquad\qquad \mathbf{g}(\mathbf{x}, \mathbf{u}, \theta) = \begin{bmatrix} \mathbf{g}_{\text{th}}(\mathbf{x}, \mathbf{u}, \theta_{\text{th}}) \\ g_{\text{el}}(\mathbf{x}, \mathbf{u}, \theta) \end{bmatrix} \tag{10}$$

The model's parameter vector is divisible as well:

$$\theta = \begin{bmatrix} \theta_{\text{th}} \\ \theta_{\text{el}} \end{bmatrix} = \begin{bmatrix} [V_{\text{ca,sm}}, V_{\text{ca,cm}}, V_{\text{ca,em}}, \dots \\ \dots V_{\text{an,sm}}, V_{\text{an,cm}}, V_{\text{an,em}}, \dots \\ \dots k_{\text{perm}}, k_{\text{cond}}, k_{\text{evap}}, \dots \\ \dots k_{\text{ca,sm,out}}, k_{\text{ca,em,in}}, k_{\text{ca,em,out}}, \dots \\ \dots k_{\text{an,sm,out}}, k_{\text{an,em,in}}, k_{\text{an,em,out}}, \dots \\ \dots k_{\text{an,leak}}, \tau_{\text{m}}]^{\text{T}} \\ \hline [\epsilon_2, R_{\text{c}}, E_{\text{ca,act}}, E_{\text{an,act}}, \dots \\ \dots K_{\text{ca}}, K_{\text{an}}, CD_{\text{ca}}, CD_{\text{an}}]^{\text{T}} \end{bmatrix} \tag{11}$$

On the one hand, the subdivision yields the thermodynamic submodel

$$\dot{\mathbf{x}} = \mathbf{f}(\mathbf{x}, \mathbf{u}, \theta_{\text{th}}), \tag{12}$$

$$\mathbf{y}_{\text{th}} = \mathbf{g}_{\text{th}}(\mathbf{x}, \mathbf{u}, \theta_{\text{th}}), \tag{13}$$

and on the other hand, it results in the electrochemical submodel

$$y_{\text{el}} = g_{\text{el}}(\mathbf{x}, \mathbf{u}, \theta). \tag{14}$$

The thermodynamic submodel (12) and (13) is only dependent on the thermodynamic parameter vector $\theta_{\text{th}}$, which also holds for the system function $\mathbf{f}$ according to Equation (9). Therefore the electrochemical parameter vector $\theta_{\text{el}}$ does not affect the states $\mathbf{x}$ and the thermodynamic outputs $\mathbf{y}_{\text{th}}$, but only the electrochemical output $y_{\text{el}}$. Thus the electrochemical submodel (14) utilizes the full parameter vector $\theta = [\theta_{\text{th}}, \theta_{\text{el}}]^{\text{T}}$. Note that the thermodynamic submodel consists of ODEs, and the electrochemical one does not. The two-step parametrization method exploits the described properties.

### 3.2. Parameter Sensitivity Analysis

The FC model used in this work has numerous unknown parameters, which raises the question of whether each parameter is unambiguously identifiable. The parameter identifiability is, in general, strongly dependent on the model structure and the available measurement data. A parameter sensitivity analysis can aid in answering these questions, and the Fisher information matrix (FIM) $\mathbf{F}$ is well-established for conducting such analysis [28]. Under the assumption of Gaussian prediction errors with zero mean values and time-independent covariances, the Cramér-Rao inequality holds [29]:

$$\text{Cov}(\theta) \succeq \mathbf{F}^{-1} \tag{15}$$

The inequality says that the inverse of $\mathbf{F}$ is the lower bound of the parameter covariances. The first step for obtaining the FIM is computing the state parameter sensitivities

$\xi_i = d\mathbf{x}/d\theta_i$, where $\theta_i$ for $i \in \{1, 2, \ldots, n_\theta\}$ denotes a parameter, and $n_\theta$ the number of parameters. They are obtainable from solving the following first-order ODE:

$$\begin{aligned}
\dot{\xi}_i &= \frac{d}{dt}\left(\frac{d\mathbf{x}}{d\theta_i}\right) = \frac{d}{d\theta_i}\left(\frac{d\mathbf{x}}{dt}\right) = \frac{d}{d\theta_i}\mathbf{f}(\mathbf{x}, \mathbf{u}, \theta) \\
&= \frac{\partial\mathbf{f}(\mathbf{x}, \mathbf{u}, \theta)}{\partial\mathbf{x}}\frac{d\mathbf{x}}{d\theta_i} + \frac{\partial\mathbf{f}(\mathbf{x}, \mathbf{u}, \theta)}{\partial\theta_i} \\
&= \frac{\partial\mathbf{f}(\mathbf{x}, \mathbf{u}, \theta)}{\partial\mathbf{x}}\xi_i + \frac{\partial\mathbf{f}(\mathbf{x}, \mathbf{u}, \theta)}{\partial\theta_i}
\end{aligned} \tag{16}$$

The second step is to calculate the output parameter sensitivities $\psi_i = d\mathbf{y}/d\theta_i$ with

$$\begin{aligned}
\psi_i &= \frac{\partial\mathbf{g}(\mathbf{x}, \mathbf{u}, \theta)}{\partial\mathbf{x}}\frac{d\mathbf{x}}{d\theta_i} + \frac{\partial\mathbf{g}(\mathbf{x}, \mathbf{u}, \theta)}{\partial\theta_i} \\
&= \frac{\partial\mathbf{g}(\mathbf{x}, \mathbf{u}, \theta)}{\partial\mathbf{x}}\xi_i + \frac{\partial\mathbf{g}(\mathbf{x}, \mathbf{u}, \theta)}{\partial\theta_i}.
\end{aligned} \tag{17}$$

The model described by Ritzberger et al. [10] has analytic derivatives available, and this work utilizes MATLAB R2020b's Symbolic Math Toolbox [30] to compute them. All output parameter sensitivities $\psi_i$ merged into one matrix yields the output parameter sensitivity matrix $\psi(t) = [\psi_1(t), \psi_2(t), \ldots, \psi_{n_\theta}(t)]$. The FIM is determinable with $\psi(t)$ taken at the sample times $t_k$ for $k \in \{0, 1, \ldots, n_k\}$, where $n_k + 1$ is the number of sample instants. Finally, using the sampled output parameter sensitivity matrix $\psi(t_k)$, the FIM is computable with

$$\mathbf{F} = \sum_{k=0}^{n_k} \psi^{\mathrm{T}}(t_k)\Sigma_{\mathbf{e}}^{-1}\psi(t_k), \tag{18}$$

where $\Sigma_{\mathbf{e}}$ denotes the prediction error covariance matrix, and $\mathbf{F} \in \mathbb{R}^{n_\theta \times n_\theta}$ holds. Under the assumption of a perfect model, the prediction error covariance $\Sigma_{\mathbf{e}}$ is identical to the measurement noise covariance and thus assumed to be known.

According to the Crámer-Rao inequality (15), the inverse of the FIM is the lower bound of the absolute parameter variances. Thus directly analyzing $\mathbf{F}$ would mean that the parameter's physical units and their magnitudes bias the analysis. Using the normalized dimensionless FIM $\mathbf{F}_{\mathrm{norm}}$ for a physical unit independent analysis resolves this issue [31–33]:

$$\mathbf{F}_{\mathrm{norm}} = \begin{bmatrix} \theta_1 & 0 & \cdots & 0 \\ 0 & \theta_2 & \ddots & \vdots \\ \vdots & \ddots & \ddots & 0 \\ 0 & \cdots & 0 & \theta_{n_\theta} \end{bmatrix} \mathbf{F} \begin{bmatrix} \theta_1 & 0 & \cdots & 0 \\ 0 & \theta_2 & \ddots & \vdots \\ \vdots & \ddots & \ddots & 0 \\ 0 & \cdots & 0 & \theta_{n_\theta} \end{bmatrix} \tag{19}$$

The identifiability of the parameters is theoretically derivable from the inverse of $\mathbf{F}_{\mathrm{norm}}$. Unfortunately, this is not always feasible because the FIM is often ill-conditioned or even singular. Using a singular value decomposition (SVD) of $\mathbf{F}_{\mathrm{norm}}$ resolves this issue and enables further analysis [32–34]:

$$\mathbf{F}_{\mathrm{norm}} = \mathbf{U}\Sigma\mathbf{V}^{\mathrm{T}} \tag{20}$$

$\mathbf{U}$ denotes the matrix containing the left singular vectors, $\Sigma = \mathrm{diag}(\sigma_1, \sigma_2, \ldots, \sigma_{n_\theta})$ the singular value matrix, and $\mathbf{V}$ the matrix containing the right singular vectors. The left and the right singular vector matrix are identical because $\mathbf{F}_{\mathrm{norm}}$ is a symmetric matrix. The analysis utilizes the right singular vector matrix $\mathbf{V} = [\mathbf{v}_1, \mathbf{v}_2, \ldots, \mathbf{v}_{n_\theta}]$, where $\mathbf{v}_l$ for $l \in \{1, 2, \ldots, n_\theta\}$ denotes a singular vector corresponding to the singular value $\sigma_l$. The singular values are interpretable as the amount of information, and the corresponding singular vectors determine the direction in the parameter space. The Euclidean norm of the vectors is 1. Thus the relative direction share of the vector component $v_{l,i}$ is $v_{l,i}^2$ [35]. The relative direction $v_{l,i}^2$ multiplied with the singular value $\sigma_l$ is the amount of information

showing into the parameter $\theta_i$'s direction. Summing up all the information of one parameter from each singular value yields the parameter's total information

$$\sigma_{\theta_i} = \sum_{l=1}^{n_\theta} v_{l,i}^2 \sigma_l. \tag{21}$$

The parameter's total information $\sigma_{\theta_i}$ indicates if a parameter is identifiable with the given model and measurement data. The most significant parameters $\theta_{\text{ms}}$ are determinable by sorting the parameters according to their total information in descending order. The most significant parameters are determined by summing up the first $n_{\theta_{\text{ms}}}$ parameters until the following inequality holds:

$$\frac{\sum_{i=1}^{n_{\theta_{\text{ms}}}} \sigma_{\theta_{\text{ms},i}}}{\sum_{i=1}^{n_\theta} \sigma_{\theta_i}} \geq \gamma \tag{22}$$

Note that $\gamma \in [0, 1]$ denotes the threshold and is adaptable for different purposes. This work uses $\gamma = 0.99999$. In this case, the most significant parameters $\theta_{\text{ms}}$ describe $\geq 99.999\%$ of $\mathbf{F}_{\text{norm}}$, and the least significant ones $\theta_{\text{ls}}$ describe $\leq 0.001\%$, which makes the latter negligible for parametrization.

*3.3. Procedure*

The proposed method parametrizes the model in two consecutive main steps:

1.  Thermodynamic submodel

    (a)  Parameter sensitivity analysis with respect to the thermodynamic parameters $\theta_{\text{th}}$ yields a subset with the most significant parameters $\theta_{\text{th,ms}}$, where $\theta_{\text{th,ms}} \subseteq \theta_{\text{th}}$ holds.
    (b)  Parametrization with respect to the most significant parameters $\theta_{\text{th,ms}}$ yields the optimized parameters $\theta_{\text{th,opt}}$. The least significant parameters $\theta_{\text{th,ls}}$ are kept constant at their initial values.

2.  Electrochemical submodel

    (a)  Solve thermodynamic submodel using the optimized thermodynamic parameters $\theta_{\text{th,opt}}$ and store the resulting model states $\mathbf{x}$ for further usage.
    (b)  Parameter sensitivity analysis with respect to the electrochemical parameters $\theta_{\text{el}}$ yields a subset with the most significant parameters $\theta_{\text{el,ms}}$, where $\theta_{\text{el,ms}} \subseteq \theta_{\text{el}}$ holds.
    (c)  Parametrization with respect to the most significant parameters $\theta_{\text{el,ms}}$ yields the optimized parameters $\theta_{\text{el,opt}}$. The least significant parameters $\theta_{\text{el,ls}}$ are kept constant at their initial values.

Combining both solutions lead to the full optimized parameter vector $\theta_{\text{opt}} = \left[ \theta_{\text{th,opt}}, \theta_{\text{el,opt}} \right]^{\text{T}}$. Assuming that the result is near the optima, the entire model with its optimized most significant parameters $\theta_{\text{ms,opt}} = \left[ \theta_{\text{th,ms,opt}}, \theta_{\text{el,ms,opt}} \right]^{\text{T}}$ can be further refined with iterative methods to approach the optima while the least significant parameters are kept constant $\theta_{\text{ls}} = [\theta_{\text{th,ls}}, \theta_{\text{el,ls}}]^{\text{T}}$ at their initial values.

The two-step parametrization method requires measurement data (according to the model's inputs and outputs), and a model separated into a submodel with ODEs and a submodel without it. The optimization goal is to minimize an objective function $J$, which contains the weighted squared errors between the simulated and measured output signals and a regularization term [19]:

$$J(\theta) = \sum_{k=0}^{n_k} (\mathbf{y}(t_k, \theta) - \mathbf{y}^*(t_k))^{\text{T}} \mathbf{Q_y} (\mathbf{y}(t_k, \theta) - \mathbf{y}^*(t_k)) + (\theta_0 - \theta)^{\text{T}} \mathbf{Q}_\theta (\theta_0 - \theta) \tag{23}$$

Here, $\mathbf{y}(t_k, \theta)$ denotes the model output at sampling instant $t_k$ for $k \in \{1, 2, \dots, n_k\}$, $\mathbf{y}^*(t_k)$ the measured output, $\mathbf{Q_y}$ the output weighting matrix, $\theta_0$ a plausible initial guess of the parameter vector, and $\mathbf{Q_\theta}$ the regularization matrix. The weighting matrix $\mathbf{Q_y}$ takes the individual weighting of each output and the different output magnitudes into account. The last term in the objective function $J$ (23), also called regularization, penalizes the deviation of the parameter vector $\theta$ from its initial guess $\theta_0$ and takes the different parameter magnitudes into account. The regularization is also beneficial if the parameters' uncertainty differs, for example, if the approximate values for some parameters are derivable from literature. The goal is to minimize the objective function $J$. In this case, the optimization problem is stated as follows:

$$\theta_{\mathrm{opt}} = \arg \min_{\theta} J(\theta)$$

$$\text{with respect to}$$

$$\theta_{i,\min} \leq \theta_i \leq \theta_{i,\max} \text{ for } i \in \{1, 2, \dots, n_\theta\}$$

(24)

Solving the optimization problem yields the optimized parameter vector $\theta_{\mathrm{opt}}$. For optimal results, the parameter space has to be constrained. The parameter bounds, $\theta_{i,\min}$ and $\theta_{i,\max}$, are derivable from physical considerations and expert knowledge.

First, parametrizing the thermodynamic submodel with its parameter vector $\theta_{\mathrm{th}}$ reduces the solution space's dimension by $\frac{8}{25} = 32\%$ compared to the global one. With the determined optimized thermodynamic parameter vector $\theta_{\mathrm{th,opt}}$ and a given input $\mathbf{u}$, the state trajectories $\mathbf{x}$ do not change anymore. Therefore, for the electrochemical submodel's parametrization, the thermodynamic submodel only needs to be solved once, and the resulting states $\mathbf{x}$ are stored for further usage. Second, parametrizing the electrochemical submodel with its parameter vector $\theta_{\mathrm{el}}$ reduces the solution space's dimension by $\frac{17}{25} = 68\%$ compared to the global one. Here only the electrochemical parameters are optimized, and the thermodynamic ones are kept constant. A parameter sensitivity analysis further reduces each sub-solution space's dimension by excluding non-significant parameters, see Section 3.4. This proceeding significantly simplifies the optimization problem. The thermodynamic submodel's parametrization solves ODEs in every iteration, and the electrochemical submodel's parametrization does not, which makes the latter considerably faster.

### 3.4. Validation of Method

This section shows the validation of the two-step parametrization method. Simulating the model with a plausible initial guess of the parameter vector $\theta_0$ and using the measured inputs $\mathbf{u}$ (described in Section 4 and depicted in Section 5 under parametrization data) yields the simulated model outputs $\mathbf{y}$. The initial values $\mathbf{x}(t_0)$ are obtainable by assuming steady-state at the first sample instant. The data sequence is appropriately selected so that at sampling instant $t_0$, the steady-state assumption holds. The simulated model outputs with additional Gaussian noise replace the measured outputs $\mathbf{y}^*$ as reference data to validate the method. In this case, the proposed method has to deliver the used parameter $\theta_0$ on average if it is an unbiased estimator.

Conducting the two-step parametrization method described in Section 3 yields the parameters' total information, shown in Figure 2a,b. Due to the floating-point arithmetic computation, the result's accuracy depends on the used hardware [36]. Therefore, the parameters with a total information value less than the hardware accuracy (indicated by the yellow line) are not identifiable. The most significant thermodynamic parameters in descending order are $k_{\mathrm{ca,em,out}}$, $k_{\mathrm{ca,sm,out}}$, $k_{\mathrm{ca,em,in}}$, $k_{\mathrm{an,sm,out}}$, $V_{\mathrm{ca,cm}}$, $k_{\mathrm{cond}}$, $k_{\mathrm{an,em,in}}$, $k_{\mathrm{perm}}$, $k_{\mathrm{an,em,out}}$, and $V_{\mathrm{an,cm}}$. The least significant ones in ascending order are $V_{\mathrm{ca,sm}}$, $V_{\mathrm{an,em}}$, $V_{\mathrm{an,sm}}$, $V_{\mathrm{ca,em}}$, $k_{\mathrm{evap}}$, $k_{\mathrm{an,leak}}$, and $\tau_{\mathrm{m}}$. Only parametrizing the most significant thermodynamic parameters reduces the dimension of the sub-solution space by $\frac{15}{25} = 60\%$ compared to the global one. The most significant electrochemical parameters in descending order are $E_{\mathrm{ca,act}}$, $E_{\mathrm{an,act}}$, $\epsilon_2$, $K_{\mathrm{ca}}$, $K_{\mathrm{an}}$, and $R_{\mathrm{c}}$. The least significant ones in ascending order

are $CD_{an}$, and $CD_{ca}$. Only parametrizing the most significant electrochemical parameters reduces the dimension of the sub-solution space by $\frac{19}{25} = 76\%$ compared to the global one. The least significant parameters are kept constant at their initial values at all times.

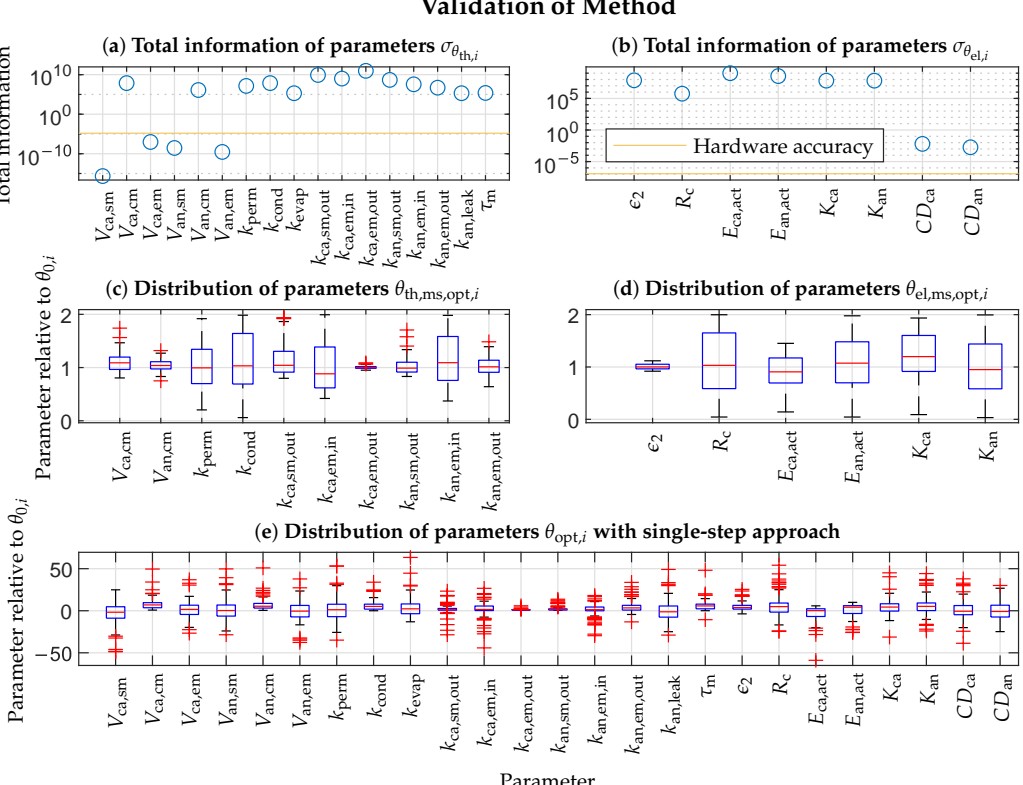

**Figure 2.** Plots (**a**,**b**): Total information of thermodynamic $\sigma_{\theta_{th,i}}$ (obtained from $\mathbf{F}_{th,norm}$) and electrochemical parameters $\sigma_{\theta_{el,i}}$ (obtained from $\mathbf{F}_{el,norm}$), respectively. Plots (**c**,**d**): Relative distribution (100 independent estimations) of optimized most significant parameters of thermodynamic $\theta_{th,ms,opt,i}$ and electrochemical submodel $\theta_{el,ms,opt,i}$, respectively. Plot (**e**): Relative distribution (100 independent estimations) of optimized parameters of entire model $\theta_{opt,i}$ with usual "single-step" approach. $\theta_{0,i}$ denotes true parameter value, and red + symbol indicates outliers in the box plots.

The two submodels with their most significant parameters are subject to parametrization. MATLAB R2020b's genetic algorithm optimizer parametrizes each submodel with respect to the most significant parameters $\theta_{ms}$ 100 times [37]. Heuristic algorithms increase the probability of finding the global minimum in a high dimensional solution space compared to iterative methods. Every iteration randomly initializes the genetic algorithm optimizer and the Gaussian noise for the model outputs. The optimizer's population size is 200, the generation limit is 20, and the remaining options remain unchanged. Figure 2c,d illustrate the relative distribution of the estimated parameters $\theta_{ms,opt}$. The conclusion is that the estimated parameters are equal to the used parameters $\theta_0$ on average, validating the proposed method. The respective spreads of the estimates correlate with the total information. The more information exists, the smaller the spread. The correlation is not perfect but will converge to the Cramer-Ráo bound (15) by increasing the population size, the generation limit, and the number of estimations.

For comparison, Figure 2e depicts the relative distribution of the estimated parameters $\theta_{opt,i}$ with a usual "single-step" approach for the entire model. The genetic algorithm optimizer minimizes the objective function $J$ (23) in a single step for the entire model without any further considerations. The resulting spreads of the parameters are much higher than those from the two-step method, which means a more complicated search for the optima.

## 4. Experimental Setup

A PEMFC system test bench was set up for the acquisition of the measurement data, which are required for the parametrization. In Figure 3, the schematic overview of the test bench is shown. The main components of the test bench are a 30 kW PEMFC stack, a hydrogen and an air supply system, the cooling circuits as well as the control system. The experimental tests were conducted by setting the load point of the FC stack via a dynamic DC/AC inverter (battery simulator). Either the current or the voltage level can be controlled by the battery simulator. Further, the battery simulator supplies the electric power, which is generated by the FC stack during operation, into the electric grid. Additionally, a power supply was designed to feed the balance of plant components (e.g., air compressor) with energy from the electric grid.

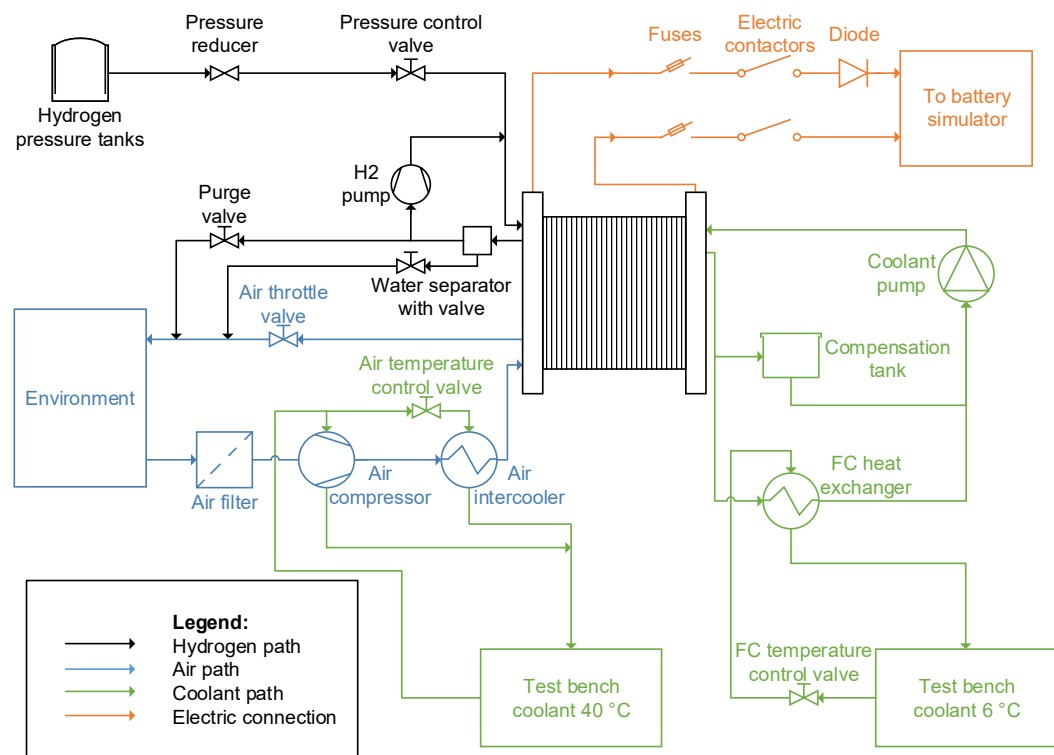

**Figure 3.** Schematic diagram of the FC test bench.

### 4.1. Media Supply

The reactants of the stack are fed by the hydrogen and air supply system. The hydrogen of high purity (99.999%) is supplied by a high pressure (300 bar) gas cylinder bundle. In the first stage, the highly compressed hydrogen is reduced to a medium pressure level of 6 bar by means of a pressure reducer. Subsequently, before the hydrogen enters the anode of the stack, the pressure controller further decompresses the hydrogen and ensures a constant pressure inside the anode. The anode of the stack operates in the flow-through mode. Therefore, a hydrogen recirculation pump is used to guarantee a constant volume flow through the anode. Besides, the stack operates without any device for humidification, either of hydrogen or air. Only the internal humidification of the reactants inside the stack is utilized. Complementary to the hydrogen purge valve, a water separator is installed at the anode side to remove the excess water periodically from the hydrogen gas. The air, which is required to supply the necessary oxygen for the electrochemical reaction, is taken in from the conditioned test bench room. For all experimental tests in this paper, the room is operated at a constant temperature of 23 °C and constant relative air humidity of 50%. The turbo compressor sucks in the ambient air from the room via an air filter, which is designed to mechanically and chemically clean the intake air from impurities. After the air compressor, the air flows through an intercooler to the cathode side of the FC stack. In

order to vary the backpressure of the stack, an electronically controlled air throttle valve is implemented at the process air exhaust of the FC stack.

### 4.2. Cooling Circuits

Two different cooling circuits of the test bench environment are used to keep the stack, the air compressor, and the air intercooler at an appropriate temperature. All the other components of the system are designed so that passive cooling is sufficient for them. The cooling circuit of the FC is thermally connected via a heat exchanger to the 6 °C cooling circuit of the test bench environment. Moreover, in the cooling circuit of the FC stack, a de-ionized coolant is used. During the operation of the FC stack, the values of the volume flow and the speed of the coolant pump are maintained constant. The 40 °C cooling circuit of the test bench environment is utilized to cool the air compressor and the intercooler. In order to guarantee a sufficient cooling of the components, flow control valves in the cooling circuits of the test bench environment are integrated. The target stack coolant inlet temperature is 55 °C. The air inlet temperature is kept at a value of 40 °C. The air compressor is supplied at all times with a constant volume flow of coolant, which is sufficient for the necessary cooling demand.

### 4.3. Test Bench Control System

Dedicated software was developed in LabVIEW and implemented on a NI CompactRio to monitor and control the FC system. With this specific software, the possibility has been created to vary each operating parameter of the FC system in an admissible range. Furthermore, with the control system the acquisition of measurement data is realized with a sampling rate of 10 Hz. In order to provide a detailed experimental investigation, a variety of sensors (temperature, pressure, mass flow, voltage, current, and humidity) are installed at all relevant positions. Further details with respect to the test bench set up are given in [38,39].

### 4.4. Experimental Tests and Operating Conditions

During the experimental tests, the load point of the stack and the air mass flow, which is controlled according to the electric current and to a constant air excess ratio of 1.5, were varied. In order to obtain measurement data under different operating states as well as operating conditions of the FC system, the load point was adjusted arbitrarily either by setting the stack voltage or the stack current level in a range, in which a stable operation of the FC is still guaranteed. All other operating parameters of the FC system (target temperatures, anode pressure setpoint, hydrogen recirculation pump speed, coolant pump speed) were kept constant under all load and operating conditions. The standard values of the constant operating parameters can be seen in Table 1. For more details regarding the experimental tests, please refer to Section 5, in which simulated and measured data are presented and discussed.

**Table 1.** Constant operating parameters and standard values of the FC system

| Operating Parameter | Value |
| --- | --- |
| Standard stack voltage range | 60–120 VDC |
| Continuous stack current | 120–400 A |
| Air compressor pressure ratio at 400 A | 1.64 (closed throttle valve) |
| Standard excess air ratio ($\lambda_{Air}$) | 1.5 |
| Air inlet temperature at cathode | 40 °C |
| Anode pressure | 1700 mbar |
| H2 pump speed | 4000 RPM |
| Stack coolant inlet temperature | 55 °C |
| Ambient temperature | 23 °C |
| Ambient pressure | 1000 mbar |
| Relative humidity of ambient air | 50% |

## 5. Results and Discussion

### 5.1. Results

This work proposes a validated two-step parametrization method, and it demonstrates the method by parametrizing the presented model using the measurement data obtained from the FC test bench. The proceeding is almost identical to the one described in Section 3.4. The differences are that the measured outputs $\mathbf{y}^*$ now serve as reference data, the optimizer's population size is 1000, and the model gets parametrized only once. Therefore all conclusions stated there still hold. Confidentiality agreements do not allow the publication of the numerical values of the optimized parameters. Figure 4a–d show the parametrization results. Simulating the model with the optimized parameter vector $\theta_{\mathrm{opt}}$ and using the measured inputs $\mathbf{u}$ described in Section 4 yields the simulated model outputs $\mathbf{y}$. The figures additionally depict the measured outputs and four model inputs as reference: the mass flow of air $\dot{m}_{\mathrm{ca,in}}$, the FC temperature $T$, the anode supply manifold pressure $p_{\mathrm{an,sm}}$, and the current $I$. The parameterized model achieves an $R^2$ value of 93% on average. For validation data, the model reaches an $R^2$ value of 91% on average (see Figure 4e–h), providing the required accuracy for control-oriented applications.

### 5.2. Discussion

The simulated outputs fit very well. However, the simulated pressures deviate more from the measurements than the voltage. The simulated anode exit manifold pressure $p_{\mathrm{an,em}}$ (Figure 4c,g) has deviations, which seem to be load-dependent. During low currents, the measured pressure is higher and vice versa. A reason could be the recirculation flow. The recirculation pump runs at a constant speed and should provide a constant flow during steady-state. The flow is not measured and is assumed constant in the model. However, steady-state conditions are not always given, especially during purging. Measuring the recirculation flow and adding it as an input to the model could minimize the deviations. The simulated cathode pressures deviate more from the validation data (Figure 4e,f) than from the parametrization data (Figure 4a,b). The maximum deviation is 5%, and a reason may be the inflow air temperature, which differs up to 10 °C between the two data sets. Incorporating the inflow air temperature into the model may improve the agreement between the simulated and measured cathode pressures. The model's simulated voltage response (Figure 4h) replicates the undershooting only in a moderate way for the validation data. One reason could be the underrepresented voltage undershooting in the parametrization data (Figure 4d). Using a data set with more often recurring voltage undershooting may improve the model's behavior in this regard. Figure 2b depicts the total information of the electrochemical parameters $\sigma_{\theta_{\mathrm{el},i}}$. The combined diffusion coefficients, $CD_{\mathrm{ca}}$ and $CD_{\mathrm{an}}$, have relatively low information. The reason is that the experiments only cover the safe operating region. The named coefficients determine the limit current and are only well identifiable in the unsafe limit current region. According to Figure 2a, the supply and exit manifold volumes ($V_{\mathrm{ca,sm}}$, $V_{\mathrm{ca,em}}$, $V_{\mathrm{an,sm}}$, and $V_{\mathrm{an,em}}$) are not identifiable. The reason is that only the pressure states needed them, which do not exist anymore. Thus the named volumes are not used in the reduced model at all. The named aspects only concern the model and the experiments, but not the parametrization method itself.

A drawback of the two-step method is that by conducting separate parameter sensitivity analyses for the two submodels, slightly different conclusions may follow compared to the complete model analysis. The reason is that the electrochemical model's output contributes additional information on the thermodynamic parameters. However, the advantage of the reduced solution space dimension compensates for this drawback. The stated conclusions derived from the FIM only hold in a region near the initial guess of the parameters $\theta_0$. Hence the FIM is only a local parameter sensitivity analysis. With another parameter vector, the conclusions from the newly derived FIM may be different. In general, parameters could be nonsignificant if the excitation is ill-suited (e.g., $CD_{\mathrm{ca}}$), it is not relevant in the model (e.g., $V_{\mathrm{ca,sm}}$), or both. High significancies are interpretable in a vice versa manner. As discussed, the parameter identifiability and the derived conclusions

are strongly dependent on the available measurement data and the model structure. With this knowledge, remodeling or a more targeted design of the experiment is possible to improve identifiability. From the engineering point of view, this information is utilizable to distribute engineering resources efficiently. Focusing on improving the significant parameter's real-world counterparts affects the system outputs substantially more than optimizing the less significant ones. Optimizing the latter will lead to hardly any changes in the output. Furthermore, more measured signals (e.g., anode and cathode outflow relative humidity, species concentrations, recirculation flow) and more "exciting" system inputs would significantly affect the obtainable results by increasing the Fisher information of the parameters, leading to better parametrization results.

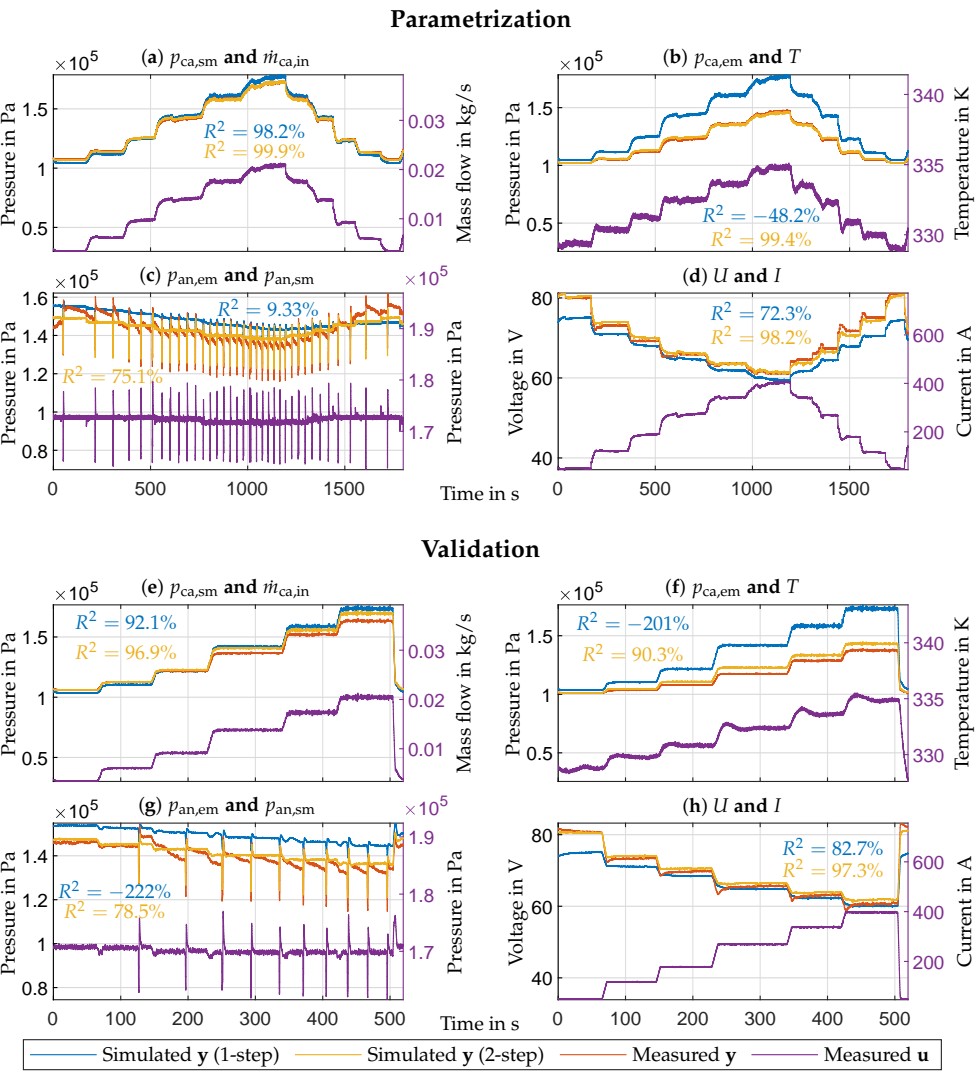

**Figure 4.** Plots (**a**,**e**): Cathode supply manifold pressure $p_{ca,sm}$ (blue, yellow and red), and air mass flow $\dot{m}_{ca,in}$ (purple). Plots (**b**,**f**): Cathode exit manifold pressure $p_{ca,em}$ (blue, yellow and red), and FC temperature $T$ (purple). Plots (**c**,**g**): Anode exit manifold pressure $p_{an,em}$ (blue, yellow and red), and anode supply manifold pressure $p_{an,sm}$ (purple). Plots (**d**,**h**): Stack voltage $U$ (blue, yellow and red), and stack current $I$ (purple). The plots depict parameterization and validation data, respectively, and **y** denotes output and **u** input.

Compared to the usual "single-step" approach, the two-step method simplifies the parametrization process significantly by reducing the solution space's dimension. As depicted in Figure 2e, the parameters resulting from a single-step parametrization of the entire model (without any further considerations) have a much larger spread, which

implicitly means a worse fit of the simulation data with the measurements. Figure 4 additionally shows the simulation results with parameters obtained from the single-step approach. The simulations achieve an average $R^2$ value of 33% for the parametrization data and $-62\%$ for the validation data, which is much worse than the proposed two-step method (93% and 91%, respectively). However, increasing the optimizer's population size and the number of generations the single-step approach would yield similar results as the two-step method. By doing so, the single-step approach searches more thoroughly through the bigger solution space of the entire model for the optima, but at the expense of exponentially growing computation time. Therefore, considering limited resources and the same optimizer options, the two-step method delivers more accurate results than the single-step approach, making the former superior in this case. A comparison of the proposed two-step method with methods consisting of more than two steps is, in this case, not meaningful because this work utilizes pre-existing data. Multiple-step methods [14,18] require appropriately designed experiments, which is usually not the case for pre-existing data. If the data allows the utilization of a method with more than two steps, then the multiple-step method will likely deliver better results than the two-step method considering similar resources. The fact that the two-step method does not need appropriately designed experiments compensates for this drawback.

## 6. Conclusions

This paper presents an efficient two-step parametrization method for FCs. A separation of the model into two submodels decreases the solution space drastically. The method parametrizes the submodels in two consecutive steps. A parameter sensitivity analysis further reduces each sub-solution space, which simplifies the search for the optima. This work demonstrates the efficient method by parametrizing an FC model with measurements and illustrates the parameter sensitivity analysis's advantages. The parameterized model's outputs replicate the validation data excellently.

The parameter sensitivity analysis is a powerful tool to determine the parameters' identifiability with a given model and measurement data. The inverse approach is also promising for future research: developing a "design of experiment controller" with a given model to maximize the parameter's identifiability in real-time.

**Author Contributions:** Conceptualization, and methodology, Z.P.D. and S.J.; investigation, data curation, Z.P.D. and C.S.; software and visualization, Z.P.D.; writing–original draft preparation, Z.P.D. and C.S.; writing–review and editing, supervision, and project administration, S.J. All authors have read and agreed to the published version of the manuscript.

**Funding:** The research leading to these results has received funding from the Mobility of the Future programme. Mobility of the Future is a research, technology and innovation funding programme of the Republic of Austria, Ministry of Climate Action. The Austrian Research Promotion Agency (FFG) has been authorised for the programme management (grant number 871503). The APC was funded by TU Wien Bibliothek.

**Institutional Review Board Statement:** Not applicable.

**Informed Consent Statement:** Not applicable.

**Data Availability Statement:** Confidentiality agreements do not allow the publication of the data presented in this study.

**Acknowledgments:** The computational results presented have been achieved, in part, using the Vienna Scientific Cluster (VSC). The authors acknowledge the financial support through Open Access Funding by TU Wien Bibliothek.

**Conflicts of Interest:** The authors declare no conflict of interest.

## Abbreviations

The following abbreviations are used in this manuscript:

| | |
|---|---|
| FC | Fuel cell |
| FIM | Fisher information matrix |
| ODE | Ordinary differential equation |
| PEMFC | Polymer electrolyte membrane fuel cell |
| SVD | Singular value decomposition |

## Nomenclature

The following symbols are used in this manuscript:

**Subscripts**

| | |
|---|---|
| 0 | Initial |
| act | Activation |
| an | Anode |
| atm | Atmosphere |
| bp | Backpressure |
| ca | Cathode |
| cm | Center manifold |
| el | Electrochemical |
| em | Exit manifold |
| $H_2$ | Hydrogen |
| in | Inflow |
| leak | Leakage |
| liq | Liquid water |
| ls | Least significant |
| max | Maximum |
| min | Minimum |
| ms | Most significant |
| m | Membrane |
| $N_2$ | Nitrogen |
| norm | Normalized |
| $O_2$ | Oxygen |
| opt | Optimized |
| out | Outflow |
| perm | Permeability |
| purge | Purge |
| reci | Recirculation |
| sm | Supply manifold |
| th | Thermodynamic |
| vap | Vapour |
| *i* | Running index for parameters |
| *k* | Sampling instant |
| *l* | Running index for singular values |

**Symbols**

| | | |
|---|---|---|
| $\alpha$ | Valve position | 1 |
| $\boldsymbol{\Psi}$ | Output parameter sensitivity matrix | $\mathbb{R}^{4 \times n_\theta}$ |
| $\boldsymbol{\psi}$ | Output parameter sensitivity vector | $\mathbb{R}^{4 \times 1}$ |
| $\boldsymbol{\Sigma}$ | Singular value matrix | $\mathbb{R}^{n_\theta \times n_\theta}$ |
| $\boldsymbol{\Sigma_e}$ | Prediction error covariance matrix | $\mathbb{R}^{4 \times 4}$ |
| $\boldsymbol{\theta}$ | Parameter vector | $\mathbb{R}^{25 \times 1}$ |
| $\boldsymbol{\theta}_{el}$ | Parameter vector of the electrochemical submodel | $\mathbb{R}^{8 \times 1}$ |
| $\boldsymbol{\theta}_{th}$ | Parameter vector of the thermodynamic submodel | $\mathbb{R}^{17 \times 1}$ |
| $\boldsymbol{\zeta}$ | State parameter sensitivity vector | $\mathbb{R}^{9 \times 1}$ |
| $\epsilon_2$ | Membrane conductivity parameter | K |
| $\gamma$ | Threshold | 1 |
| $\lambda_{Air}$ | Excess air ratio | 1 |
| $\mathbf{F}$ | Fisher information matrix | $\mathbb{R}^{n_\theta \times n_\theta}$ |
| $\mathbf{f}$ | System function of the reduced model | $\mathbb{R}^{9 \times 1}$ |
| $\mathbf{f}_{nr}$ | System function of the non-reduced model | $\mathbb{R}^{12 \times 1}$ |
| $\mathbf{f}_{th}$ | System function of the thermodynamic submodel | $\mathbb{R}^{9 \times 1}$ |
| $\mathbf{g}$ | Output function of the reduced model | $\mathbb{R}^{4 \times 1}$ |
| $\mathbf{g}_{nr}$ | Output function of the non-reduced model | $\mathbb{R}^{4 \times 1}$ |
| $\mathbf{g}_{th}$ | Output function of the thermodynamic submodel | $\mathbb{R}^{3 \times 1}$ |
| $\mathbf{Q_y}$ | Output weighting matrix | $\mathbb{R}^{4 \times 4}$ |
| $\mathbf{Q_\theta}$ | Regularization matrix | $\mathbb{R}^{25 \times 25}$ |
| $\mathbf{U}$ | Left singular vector matrix | $\mathbb{R}^{n_\theta \times n_\theta}$ |
| $\mathbf{u}$ | Input vector | $\mathbb{R}^{8 \times 1}$ |
| $\mathbf{V}$ | Right singular vector matrix | $\mathbb{R}^{n_\theta \times n_\theta}$ |

| | | |
|---|---|---|
| **v** | Right singular vector | $\mathbb{R}^{n_\theta \times 1}$ |
| **x** | State vector of the reduced model | $\mathbb{R}^{9 \times 1}$ |
| **x**$_{\text{nr}}$ | State vector of the non-reduced model | $\mathbb{R}^{12 \times 1}$ |
| **x**$_{\text{th}}$ | State vector of the thermodynamic submodel | $\mathbb{R}^{9 \times 1}$ |
| **y** | Output vector | $\mathbb{R}^{4 \times 1}$ |
| **y**$^*$ | Measured output vector | $\mathbb{R}^{4 \times 1}$ |
| **y**$_{\text{th}}$ | Output vector of the thermodynamic submodel | $\mathbb{R}^{3 \times 1}$ |
| $\sigma$ | Singular value | |
| $\sigma_{\theta_i}$ | Total information of parameter $\theta_i$ | |
| $\tau$ | Time constant | s |
| $\theta$ | Parameter | |
| $\varphi$ | Relative humidity | 1 |
| $a$ | Water activity | 1 |
| $CD$ | Combined diffusion coefficient | mol/s |
| $E$ | Energy | J |
| $g_{\text{el}}$ | Output function of the electrochemical submodel | $\mathbb{R}^{1 \times 1}$ |
| $I$ | Current | A |
| $J$ | Objective function | $\mathbb{R}^{1 \times 1}$ |
| $K$ | Intrinsic exchange current parameter | A/m$^2$ |
| $k$ | Nozzle or mass flow coefficient | kg/(s · Pa) |
| $k_{\text{cond}}$ | Condensation coefficient | 1/s |
| $k_{\text{evap}}$ | Evaporation coefficient | 1/(s · Pa) |
| $m$ | Mass | kg |
| $n_k$ | Number of sample instants ($n_k + 1$) | 1 |
| $n_\theta$ | Number of parameters | 1 |
| $p$ | Pressure | Pa |
| $R$ | Mass-specific gas constant | J/(kg · K) |
| $R_{\text{c}}$ | Ohmic contact resistance | Ω |
| $T$ | Fuel cell temperature | K |
| $t$ | Time | s |
| $U$ | Voltage | V |
| $V$ | Volume | m$^3$ |
| $v$ | Right singular vector component | |
| $y_{\text{el}}$ | Output of the electrochemical submodel | $\mathbb{R}^{1 \times 1}$ |

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
