# Peer review of "Efficient Two-Step Parametrization of a Control-Oriented Zero-Dimensional Polymer Electrolyte Membrane Fuel Cell Model Based on Measured Stack Data"

_processes, doi:10.3390/pr9040713_

Round 1
Reviewer 1 Report
This paper presented a two-step parametrization of a control-oriented zero-dimensional polymer electrolyte membrane fuel cell model. Subdividing the parametrization into two consecutive subproblems reduced the solution space significantly. A parameter sensitivity analysis further reduced each sub-solution space by excluding non-significant parameters. The review comments for this paper are as follows:
- Line 13, please check the journal author guidelines for the number of the keywords. It looks too many keywords in the current paper.
- What are the main advances of the polymer electrolyte membrane fuel cells? Please compare it with other materials used in fuel cells in Introduction.
- Line 45, the authors claimed that this work presented an efficient two-step parametrization. How did you improve in this two-step method compared with other steps parametrization? The authors did not explain the reasons to choose this method.
- Line 140, what is the subject for "describes the nonlinear FC state-space model"? And also, Lines 171, 184, 224, 233, 234, 235, 281.
- Was the two-step method more accurate than one-step and more than two steps during the parametrization?
- It is better to adjust the words in Figure 3.
- In Section 7, more results from the simulation and experiments should be added. What are the major influencing factors effect on the system efficiency? It is better to compare the results obtained from this work with single-step method. How is the accuracy improvement?
Reviewer 2 Report
In the manuscript “Efficient two-step parametrization of a control-oriented zero-dimensional polymer electrolyte membrane fuel cell model based on measured stack data” submitted for publication in the journal Processes, the authors propose a new efficient two-step method for parametrizing control-oriented zero-dimensional physical polymer electrolyte membrane fuel cell models with measured stack data. It covers the numerical results of the theoretical model in MATLAB and is validated by an identification experiment. However, I have some comments on the interpretation of the results. Therefore, I believe this manuscript needs revision, addressing the points will certainly increase the value of this paper.
- Why the machine precision at Figure 2b is quite off the parameters’ total information.
- An extended explanation should be added to the manuscript for figure 2c and 2e which may help the reader to understand why some of the parameters are off the precision.
Round 2
Reviewer 1 Report
Please try to reduce the sections of this research article. Some sections should be merged.
